# Longitudinal Analysis of Coronavirus-Neutralizing Activity in COVID-19 Patients

**DOI:** 10.3390/v14050882

**Published:** 2022-04-23

**Authors:** Florian D. Hastert, Lisa Henss, Christine von Rhein, Julia Gerbeth, Imke Wieters, Frauke Borgans, Yascha Khodamoradi, Kai Zacharowski, Gernot Rohde, Maria J.G.T. Vehreschild, Barbara S. Schnierle

**Affiliations:** 1Department of Virology, Paul-Ehrlich-Institut, Paul-Ehrlich Strass 51-59, 63225 Langen, Germany; florian.hastert@pei.de (F.D.H.); lisa.henss@pei.de (L.H.); christine.vonrhein@pei.de (C.v.R.); julia.gerberth@pei.de (J.G.); 2Zentrum für Innere Medizin, Infektiologie, Universitätsklinikum Frankfurt, Goethe Universität Frankfurt, Theodor-Stern-Kai 7, 60590 Frankfurt am Main, Germany; imke.wieters@kgu.de (I.W.); frauke.borgans@kgu.de (F.B.); yascha.khodamoradi@kgu.de (Y.K.); maria.vehreschild@kgu.de (M.J.G.T.V.); 3Klinik für Anästhesiologie, Intensivmedizin und Schmerztherapie, Universitätsklinikum Frankfurt, Goethe Universität, Theodor-Stern-Kai 7, 60590 Frankfurt am Main, Germany; kai.zacharowski@kgu.de; 4Medizinische Klinik 1, Pneumologie/Allergologie, Universitätsklinikum Frankfurt, Goethe-Universität, Theodor-Stern-Kai 7, 60590 Frankfurt am Main, Germany; gernot.rohde@kgu.de

**Keywords:** SARS-CoV-2, Delta, Omicron, SARS-CoV-1, NL63, neutralization, vaccination

## Abstract

The severe acute respiratory syndrome coronavirus-2 (SARS-CoV-2) pandemic has now been continuing for more than two years. The infection causes COVID-19, a disease of the respiratory and cardiovascular system of variable severity. Here, the humoral immune response of 80 COVID-19 patients from the University Hospital Frankfurt/Main, Germany, was characterized longitudinally. The SARS-CoV-2 neutralization activity of serum waned over time. The neutralizing potential of serum directed towards the human alpha-coronavirus NL-63 (NL63) also waned, indicating that no cross-priming against alpha-coronaviruses occurred. A subset of the recovered patients (*n* = 13) was additionally vaccinated with the mRNA vaccine Comirnaty. Vaccination increased neutralization activity against SARS-CoV-2 wild-type (WT), Delta, and Omicron, although Omicron-specific neutralization was not detectable prior to vaccination. In addition, the vaccination induced neutralizing antibodies against the more distantly related SARS-CoV-1 but not against NL63. The results indicate that although SARS-CoV-2 humoral immune responses induced by infection wane, vaccination induces a broad neutralizing activity against multiple SARS-CoVs, but not to the common cold alpha-coronavirus NL63.

## 1. Introduction

The severe acute respiratory syndrome coronavirus-2 (SARS-CoV-2) belongs to the *Coronaviridae* family and is the causative agent of coronavirus disease-2019 (COVID-19), which first emerged in the Hubei province in China [1]. The virus rapidly spread worldwide, and the World Health Organization (WHO) declared COVID-19 a pandemic on 11 March 2020. Coronaviruses can cause different diseases in humans. There are four endemic human coronaviruses (huCoV); OC43 and 229E belong to the beta coronavirus family along with SARS-CoV-2, while HKU1 and NL63 are alpha-coronaviruses. They are the causative agents of common colds. In contrast, two other coronaviruses, severe acute respiratory syndrome virus (SARS-CoV-1) and Middle East respiratory syndrome virus (MERS-CoV), have high pathogenic potential with 15–30% mortality in humans and have caused small epidemics of severe pneumonia [2]. COVID-19 disease severity is variable and ranges from asymptomatic up to life-threatening pneumonia with severe respiratory distress [3,4].

The coronavirus structural proteins, the surface glycoprotein termed spike (S) and the more abundant nucleocapsid (N), are the principal immunogens used for the detection of anti-SARS-CoV-2-specific antibodies [5]. The spike protein consists of two subunits S1 and S2. S1 mediates the attachment of the virus to human cells via its receptor-binding domain (RBD), and S2 mediates the fusion of the viral and cellular membranes. Antibodies that bind to the spike protein, and in particular to the RBD, can directly neutralize coronaviruses [6]. Therefore, the spike protein is the major immunogen used in vaccines to induce a SARS-CoV-2-specific immune response [6].

After more than two years of the pandemic, the pathogenesis of SARS-CoV-2 is still not well understood. Longitudinal analyses of humoral immune responses are currently being carried out to determine the protective ability of antibodies [7,8,9]. Here, we analyzed the humoral immune response of COVID-19 patients in Germany over a period of more than 400 days after their positive PCR test.

The SARS-CoV-2 pandemic has resulted in the development of virus variants of concern (VOC), with the latest being Delta and Omicron. Omicron has a large number of amino acid substitutions, insertions, and deletions in the viral spike protein compared to the wild-type (WT) and apparently reflects the evolution of a new SARS-CoV-2 serotype [10]. Here, we tested samples of a subcohort of convalescent patients after vaccination with the mRNA vaccine Comirnaty for a booster response against the SARS-CoV-2 WT and the variants Delta and Omicron as well as the more distantly related SARS-CoV-1 and the common cold inducing huCoV-NL63

This longitudinal analysis aims to help to characterize the immune response and support the identification of correlates of protection needed for the development of vaccines, vaccine booster doses, and therapeutic antibodies.

## 2. Materials and Methods

### 2.1. Cell Culture

HEK293T (American Type Culture Collection CRL-3216) and HEK293T-hACE2 cells [11] were grown in Dulbecco’s modified Eagle medium (DMEM; Sigma, Taufkirchen, Germany) supplemented with 10% fetal bovine serum (Sigma, Taufkirchen, Germany), 5% L-glutamine (200 mM; Lonza, Verviers, Belgium), and 1% penicillin/streptavidin (Fisher Scientific, Schwerte, Germany) at 37 °C under 5% CO_2_. Medium for HEK293T-hACE2 cells was additionally supplemented with 50 µg/mL Zeocin™ (Fisher Scientific, Schwerte, Germany).

### 2.2. Patient Serum Samples

Human naïve serum from volunteer blood donors was obtained from the German Red Cross and was collected prior to the introduction of SARS-CoV-2 into Germany. Human sera from SARS-CoV-2 PCR-positive patients were obtained from the University Hospital Frankfurt/Main under “COVID Capnetz” ethical approval (#11/17). The study was reviewed and approved by the ethics committee of Frankfurt University. The samples were taken from multiple donations from single patients. Blood samples were collected between 14 and 537 days after PCR confirmation of infection. The initial cohort comprised 51 female blood donors aged from 19 to 75 years and 29 male blood donors aged from 18 to 65 years. Of the female subcohort, 44 individuals were assigned clinical scores of 1 or 2, and 7 individuals with more severe symptoms were assigned clinical scores between 3 and 6 [12]. For the male subgroup, 25 mild symptomatic and 4 hospitalized individuals with clinical scores between 3 and 6 were enrolled in this study. Blood samples taken after full vaccination with the Comirnaty mRNA vaccine were available for a subgroup of patients comprising 9 female and 4 male individuals. Of these, 9 had received one dose of Comirnaty, and 4 had received 2 doses, which was considered a full vaccination at that time.

### 2.3. Serum Neutralization Assay Using Pseudotyped Lentiviral Vectors

Lentiviral vectors were produced in HEK293T cells by co-transfection using Lipofectamine^®^ 2000 (Thermo Fisher, Darmstadt, Germany) as described previously [13]. Plasmids encoding HIV-1 gag/pol, rev, the luciferase-encoding lentiviral vector genome, and the spike gene from SARS-CoV-2 wild-type 614D (MN908947), SARS-CoV-2 Delta (OK091006.1), SARS-CoV-2 Omicron BA.1 (OM287553.1), SARS-CoV-1 (AY278741.1) or huCoV-NL63 (AFV53148.1) were transfected. Spike genes that lacked the last 19 carboxy-terminal amino acids were synthesized (Eurofins, Ebersberg, Germany; IDT, Leuven, Belgium) and cloned into the vector pcDNA 3.1^(+)^ as described before [14]. After harvest, vector particles were concentrated by ultracentrifugation and stored at −80 °C or for Omicron pseudotypes, were used directly for neutralization assays. Pseudotyped vectors and serially diluted human serum (from 1:60 to 1:14,580) were incubated for 45 min at 37 °C and used to transduce HEK293T-hACE2 cells in triplicate [14]. After 48 h, britelite plus luciferase substrate (PerkinElmer, Waltham, MA, USA) was added to quantify luciferase activity. The reciprocal area under the curve (1/AUC) value that was calculated for each sample corresponds to the respective neutralization activity. Mean 1/AUC values from 12 previously SARS-CoV-2-naïve individuals plus two standard deviations were used as background values.

### 2.4. Statistical Analysis of the Neutralization Experiments and Software

AUC values were determined using the GraphPad Prism 7.04 software (La Jolla, San Diego, CA, USA). Longitudinal development of neutralization was analyzed and plotted with RStudio Version 1.2.1335 (https://www.R-project.org/ (accessed on 17 March 2022)). Half-lives (t_1/2_) were calculated using a simple linear regression model with RStudio 1.2.1335, correlating the reciprocal AUC values with days after the positive PCR test. Slopes of the linear regression trendlines were calculated with RStudio 1.2.1335.

Sequences of CoV-receptor-binding domains (RBDs) were compared with ClustalOmega [15], and a neighbor-joining tree was calculated and visualized with JalView [16].

Significance is indicated with three stars (***) for a *p*-value ≤ 0.001, two stars (**) for a *p*-value ≤ 0.01, and one star (*) denotes a *p*-value ≤ 0.05; NS denotes samples that are not statistically different.

## 3. Results

### 3.1. Characterization of COVID-19 Patients

Initial and follow-up serum samples were collected from 80 SARS-CoV-2 PCR-positive patients with clinical scores ranging between 1 and 6, with scores 1 and 2 reflecting mild symptoms and 3–6 hospitalized patients. The serum samples from the 80 patients (51 female and 29 male) were collected between 9 March 2020 and 15 September 2021 in Frankfurt/Main, Germany. Patients had been infected with SARS-CoV-2 between 5 March and 14 July 2020; at this time point, the original Wuhan strain with the D614G mutation was circulating in Germany. The patients were from 18 to 75 years old, and initial samples were collected at different times between days 14 and 269 after confirmation of infection by PCR. Most patients had mild disease (86.25%), and 13.75% of the patients had severe disease.

### 3.2. SARS-CoV-2 WT and NL63 Neutralization Activity of Serum from Convalescent Patients Wanes over Time

The humoral immune response against SARS-CoV-2 was studied by determining the ability of serum to neutralize SARS-CoV-2 WT (D614) using lentiviral vectors pseudotyped with the SARS-CoV-2 spike protein [14]. The neutralization activity of serum was determined by analyzing the AUC of the resulting luciferase generated light signals plotted against the serum dilutions. SARS-CoV-2 WT-neutralizing activity was highest early after infection and decreased over time (Figure 1A). Overall, the neutralization activity of patient-derived antibodies was quite diverse but decreased significantly (*p* < 0.0001) over time, and the mean neutralization half-life was 140 days.

Most people have preexisting immune responses against the huCoVs that cause common colds [14,17,18]; hence, we also studied the longitudinal NL63-neutralization potential in serum from COVID-19 patients. Neutralization activity was determined with lentiviral vectors pseudotyped with the NL63 spike protein and is represented as the reciprocal AUC (Figure 1B). The neutralization activity of patient-derived antibodies was again quite diverse. Interestingly, the NL63 neutralization activity decreased significantly (*p* < 0.0005) over time with a half-life of 218 days, slightly longer than that for SARS-CoV-2. Background levels could not be determined, as, due to the high seroprevalence, NL63 negative serum was not available.

### 3.3. The Impact of mRNA Vaccination on Coronavirus-Specific Neutralization in Recovered COVID-19 Patients

A subgroup of recovered COVID-19 patients who had provided blood samples after vaccination with the mRNA vaccine Comirnaty was available. Serum from these individuals was first analyzed for neutralization activity against SARS-CoV-2 WT, Delta, and Omicron. The serum samples were divided into three groups, early after infection (52 ± 6 days), late after infection (274 ± 22 days), and after vaccination (35 ± 10 days). The neutralizing activity against SARS-CoV-2 WT and Delta waned over time, as shown above. However, vaccination increased the neutralization activity 31.5-fold against WT and 11.1-fold against Delta compared to the titer early after the infection (Figure 2). Overall, the neutralization values for Delta and WT were almost identical at early and late time points after infection, with mean reciprocal AUCs of 1.33 and 0.51 for Delta and 1.13 and 0.67 for WT, and were significantly different (*p* < 0.005) only after vaccination with mean AUC values of 5.27 for Delta and 22.06 for WT.

Serum from recovered patients had no neutralizing activity against the Omicron variant, but the Comirnaty vaccination increased the neutralization activity against Omicron to detectable levels in half of the patients (Figure 2), although the mean reciprocal AUC of 0.98 was very low.

This analysis was extended to more distantly related coronaviruses. The SARS-CoV-1 spike protein has 76.06% identity to that of SARS-CoV-2 WT, while the NL63 spike has only 31.27%, the Delta and Omicron spikes have 99.20% and 96.98% identity, respectively (Figure 3A). Neutralization of SARS-CoV-1 and the common cold coronavirus NL63 was assessed as before with pseudotyped vectors. Before vaccination, recovered patients had low or background levels of SARS-CoV-1-neutralizing antibodies early after the SARS-CoV-2 infection (days 52 ± 6). Vaccination significantly boosted the induction of SARS-CoV-1-neutralizing antibodies (4.8-fold) in all but one recovered patient, with mean values starting at 0.385 and reaching 1.85 after the vaccination. High levels of neutralizing titers were reached for individual patients (Figure 3B).

In contrast, NL63-specific neutralizing antibodies were not increased by the vaccination, and titers declined further (Figure 3C).

## 4. Discussion

Here, longitudinal serological analyses of 80 COVID-19 patients from the University Hospital Frankfurt/Main were performed. We observed that SARS-CoV-2 WT-specific neutralization activity declined with a half-life of 140 days. This is in agreement with previous results described by others [7,8,19,20,21] and mirrors the vaccination recommendations currently active, which propose the vaccination of recovered COVID-19 patients 3–6 months after infection to ensure protection.

When neutralization of the human alpha-coronavirus NL63 was analyzed, we found that it followed a similar kinetic to SARS-CoV-2 WT-specific antibodies. NL63 reactivity of human serum is very common, and neutralizing antibodies were detected in all sera tested. COVID-19 patients showed a decline in NL63-neutralizing antibodies over time, which might reflect the absence of seasonal reinfections due to mask wearing. Cross-reactive IgG responses towards beta-CoV have been described, but in that study, alpha-CoV-specific IgG was not induced in patients, confirming our results [22]. The data would argue against an original antigenic sin (OAS) response to a SARS-CoV-2 infection after NL63 exposure. OAS is the induction of antibodies directed against an original, priming antigen rather than a similar boosting antigen. Since prior exposure to huCoVs is very common, there are concerns that preexisting antibodies against huCoVs might lead to aberrant immune responses against SARS-CoV-2. Recently, OAS was observed in a mouse model for heterologous beta coronaviruses but not in children following SARS-CoV-2 infections [23]. However, here and also in other studies, no cross-booster effects were observed [24]. In addition, in the subgroup of COVID-19 patients that were vaccinated, no increase in NL63 neutralization activity was detected after the vaccination. Although, it cannot be excluded that a slight increase in NL63 neutralization might have occurred that slowed the decline of antibody levels. Hence, strong cross-reactions against NL63 are not likely.

During the SARS-CoV-2 pandemic, several variants of concern have developed. Partial immune escape from convalescent patient or vaccinee sera has been described for the Beta, Gamma, and Delta variants [25,26,27]. In addition, the Omicron variant has caused high numbers of reinfections of convalescent and vaccinated individuals [28]. Omicron is currently globally dominant and has replaced the previously dominant Delta variant.

The spike protein mediates virus cell entry, and virus transmission and immune evasion mutations mainly occur in this protein. Fifteen changes in the Omicron spike compared to the Wuhan Hu-1 strain are located in the ACE2 receptor-binding domain (RBD) and twelve in the N-terminal domain (NTD) [29]. Polyclonal neutralizing antibody responses target exactly these regions [30,31], and levels of neutralizing antibodies are predictive of immune protection from symptomatic SARS-CoV-2 infection [32]. Therefore, the spike protein is the antigen used to induce an immune response by vaccination. Vaccination of recovered patients restored and increased the neutralizing activity to all tested variants and even to Omicron, which was resistant to serum from recovered patients. The vaccination improved the quality of immune responses, which also extended to cross-neutralization of the related SARS-CoV-1, which shares only 73% sequence identity in its RBD with SARS-CoV-2 WT. Neutralization activity against SARS-CoV-1 has also been detected in mRNA-vaccine-boosted individuals [33,34]. This suggests that a SARS-CoV-2 infection followed by vaccination gives a similar breadth of antibody response as three vaccinations. The RBD of NL63 has no significant similarities with the SARS-CoV-2-WT, which might well explain why no noteworthy increase in neutralization ofNL63 after SARS-CoV-2 infection or vaccination was observed.

Although the Omicron RBD shares about 93% identity with the SARS-CoV-2 WT, no serum neutralization was detected in convalescent patients. Similar results have been obtained for vaccinated individuals or monoclonal antibodies [35,36,37,38,39,40,41]. This is readily explained by the high number of amino acids alterations in the Omicron RBD that directly interact with the ACE2 receptor [29]. Interestingly, although the SARS-CoV-1 RBD has far less similarity to the SARS-CoV-2 WT than Omicron, it still possesses similar amino acids in 7 out of 15 positions that are mutated in Omicron, and only 1 of these 15 is homologous to Omicron. This might explain the observed increase in SARS-CoV-1 neutralization after vaccination.

Vaccine development mainly relies on the assumption that antibodies will be essential for protection as deduced from animal models [42,43]. However, the minimum threshold level for protection is still unknown. Breakthrough infections in convalescent patients will give valuable insights into the antibody levels or T-cell responses able to provide protection. Although Omicron is hardly neutralized in vitro, most reinfected individuals appear to have only mild disease symptoms, indicating that other mechanisms contribute to protection from severe diseases, such as cellular immune responses, which have been described to be highly conserved [44,45,46,47], or Fc-receptor-dependent responses, which are also conserved against Omicron [48].

Several heterologous vaccines are currently in development with the aim of extending the neutralization activity to Omicron or newly emerging Omicron variants. Recently, it was shown that sera from unvaccinated, naïve individuals who became infected with Omicron could mainly neutralize Omicron but not other VOCs. An Omicron-specific vaccination or infection might not be sufficient to protect against an infection with other SARS-CoV-2 variants [49]. For full protection, antibodies against SARS-CoV-2 WT and Omicron might be needed. However, a boost with an Omicron-specific mRNA vaccine after a WT mRNA vaccination did not provide enhanced immune responses or protection compared to a WT-specific boost in non-human primates [50]. On the other hand, a chimeric mRNA vaccine encoding a chimera of the Delta-RBD variant and the entire S antigen of Omicron induced potent and broadly neutralizing antibodies against Omicron (both BA.1 and BA.2) and Delta in mice [51]. Future heterologous vaccines might be superior if they induce immune responses against both SARS-CoV-2 WT and highly mutated VOC such as Omicron.

## Figures and Tables

**Figure 1 viruses-14-00882-f001:**
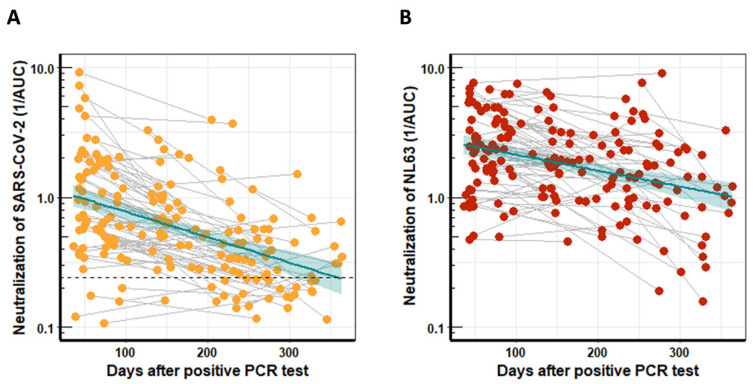
Neutralization activity of longitudinal serum samples from COVID-19 patients. The neutralization activity was determined with pseudotyped lentiviral vectors and calculated using the reciprocal areas under the curve (1/AUC). (**A**) SARS-CoV-2 WT neutralization activity of serum from 80 COVID-19 patients who had provided two blood donations, plotted against time after PCR-confirmed COVID-19 infection. The dashed line indicates the mean 1/AUC background values of the assay, which were generated from sera of 12 previously SARS-CoV-2 naïve individuals plus two standard deviations. (**B**) Serum neutralization of NL63 for the same cohort as in (**A**) plotted against time after PCR-confirmed COVID-19 infection. The turquoise line indicates the linear regression trendline, and shaded areas represent the respective standard derivations.

**Figure 2 viruses-14-00882-f002:**
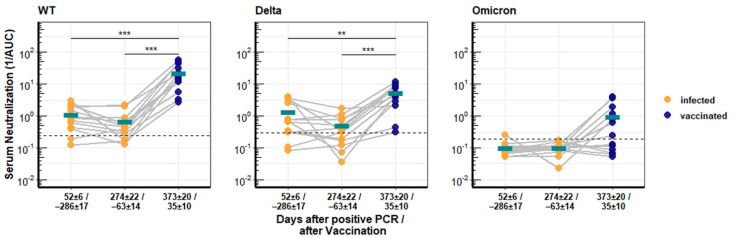
Longitudinal neutralization activity of serum from COVID-19 patients against different SARS-CoV-2 variants of concern. Antibody titers from 13 patients with PCR-confirmed SARS-CoV-2 infection and after vaccination with Comirnaty. Neutralization activity against SARS-CoV-2 WT, Delta, and Omicron was determined with pseudotyped lentiviral vectors and is depicted as reciprocal area under the curve. The turquoise bars give the respective arithmetic means of all samples, and asterisks indicate significant changes. Significance is indicated with three stars (***) for a *p*-value ≤ 0.001 and two stars (**) for a *p*-value ≤ 0.01.

**Figure 3 viruses-14-00882-f003:**
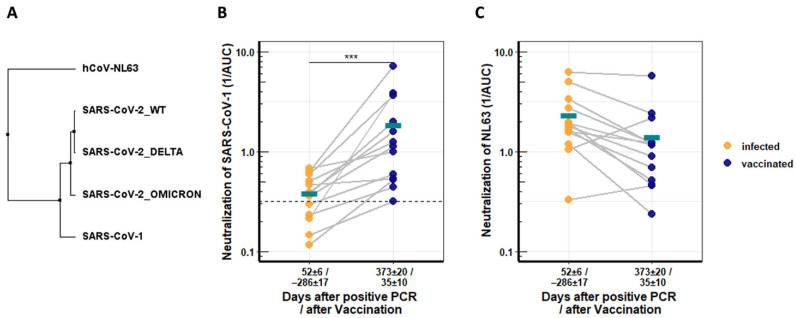
Longitudinal neutralization activity from serum of COVID-19 patients against distantly related CoVs. (**A**) Neighbor-joining tree based on receptor binding domains (RBDs) of different coronaviruses. (**B**) Neutralization activity of COVID-19 convalescent serum against SARS-CoV-1 before and after Comirnaty vaccination. (**C**) Neutralization activity of COVID-19 convalescents serum against NL63 before and after Comirnaty vaccination. Serum neutralization was determined with pseudotyped lentiviral vector particles. Asterisks indicate significance levels: (***) for a *p*-value ≤ 0.001, and turquoise bars denote respective mean values.

## Data Availability

The data presented in this study are available on request from the corresponding author. The data are not publicly available due to ethical restrictions.

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
