# Peer review of "Longitudinal Analysis of Coronavirus-Neutralizing Activity in COVID-19 Patients"

_viruses, 2022, doi:10.3390/v14050882_

Round 1

Reviewer 1 Report

Hastert et al. evaluated the human coronavirus neutralization potential of longitudinal samples collected from a primarily convalescent cohort. In addition to SARS-CoV-2 neutralization, the authors also measured activity against the endemic NL63 coronavirus. The half-lives of CoV-2 and NL63 neutralization were 140 and 218 days, respectively.

This manuscript presents a relatively simple but valuable survey of serological responses in a moderately sized cohort of subject. In addition to the direct utility of the SARS-CoV-2 data, the authors' inclusion of an endemic coronavirus (NL63) addresses important considerations surrounding what effect priming by these seasonal infections could have.

The manuscript is well organized in its current form; however, the authors may wish to consider a few minor points that could aid in understanding:

  • In its current pre-publication form, the shaded band in figure 1 is quite faint and difficult to see; also, the legend lists it as representing "standard derivation"
  • The figure 1 legend may also benefit from reiterating the source of the dashed line that's stated in the methods
  • On line 183, a reciprocal AUC value of 0.98 is referenced - is this a mean value that's limited to the subjects who measured above background for omicron response? The corresponding mean bar in figure 2 appears lower. If so, this should be made clearer in the text.

Author Response

Reviewer 1

Hastert et al. evaluated the human coronavirus neutralization potential of longitudinal samples collected from a primarily convalescent cohort. In addition to SARS-CoV-2 neutralization, the authors also measured activity against the endemic NL63 coronavirus. The half-lives of CoV-2 and NL63 neutralization were 140 and 218 days, respectively.

This manuscript presents a relatively simple but valuable survey of serological responses in a moderately sized cohort of subject. In addition to the direct utility of the SARS-CoV-2 data, the authors' inclusion of an endemic coronavirus (NL63) addresses important considerations surrounding what effect priming by these seasonal infections could have.

The manuscript is well organized in its current form; however, the authors may wish to consider a few minor points that could aid in understanding:

  • In its current pre-publication form, the shaded band in figure 1 is quite faint and difficult to see; also, the legend lists it as representing "standard derivation"

As suggested by the reviewer, the shaded band in Figure 1 was changed to a darker color.

  • The figure 1 legend may also benefit from reiterating the source of the dashed line that's stated in the methods

A sentence was added in the figure legend of Figure 1. “The dashed line indicates the mean 1/AUC background values of the assay, which were generated from sera of 12 previously SARS-CoV-2 naïve individuals plus two standard deviations”

  • On line 183, a reciprocal AUC value of 0.98 is referenced - is this a mean value that's limited to the subjects who measured above background for omicron response? The corresponding mean bar in figure 2 appears lower. If so, this should be made clearer in the text.

A value of 0.98 is mentioned only in line 209 and as stated in the text, it represents a mean value of all samples. We included this information in the figure legend in line 216.

However, there is a mistake with the bars indicating all the mean values. Unfortunately, the bar was not changed, when we converted the 1/AUC values into a logarithmic scale. We changed this now in Figure 2 and Figure 3. Thank you very much for finding this mistake.

In addition, we added a missing doi number to the references.

Reviewer 2 Report

Authors showed the direct data of neutralization activities to several variant comparing alpha CoV NL63 - is that the only coronavirus associated with common cold? or that is an example which has no similarities in RBD.

In manuscript, there are mixed names of NL63 with and without hCoV or alpha CoV... 

page line 89 In methods, there are descriptions of clinical scores without references or explanations for calculation.

Additionally, is there no relationship between clinical score and immunogenicity?

Author Response

Reviewer 2

  • Authors showed the direct data of neutralization activities to several variant comparing alpha CoV NL63 - is that the only coronavirus associated with common cold? or that is an example which has no similarities in RBD.

As stated in the introduction, there are four endemic human coronaviruses (huCoV); OC43 and 229E, belong to the beta coronavirus family along with SARS-CoV-2, while HKU1 and NL63 are alpha coronaviruses. We chose NL63, because it also uses ACE2 as a receptor for virus entry and might therefore interfere with SARS-CoV-2 immunity. However, as we showed here, this is not the case.

  • In manuscript, there are mixed names of NL63 with and without hCoV or alpha CoV... 

The human endemic coronavirus NL63 is now named NL63 all over the manuscript. However, the word alpha CoV is a more general term which decribes alpha coronaviruses and we kept this name.

  • page line 89 In methods, there are descriptions of clinical scores without references or explanations for calculation.

We are sorry, but we forgot the reference, which we included now in the text in line 104.

  • Additionally, is there no relationship between clinical score and immunogenicity?

In our initial analyses of COVID-19 patients, we saw a clear correlation of clinical score and antibody levels [1]. However here, we did not observe a correlation, because the sera were mainly from donors, which required only minimal care and were not severely symptomatic. The ratio was 69 patients with score 1-2 and 11 patients with score 3-6. Therefore this type of analysis could not be performed.

In addition, we added a missing doi number to the references.
